# Mechanisms of Cell–Cell Fusion in SARS-CoV-2: An Evolving Strategy for Transmission and Immune Evasion

**DOI:** 10.3390/v17111405

**Published:** 2025-10-22

**Authors:** Kate Chander Chiang, Cheng En Nicole Chiu, Mazharul Altaf, Mark Tsz Kin Cheng, Ravindra K. Gupta

**Affiliations:** 1University College Dublin School of Medicine, Belfield, 4 Dublin, Ireland; kate.chiang@ucdconnect.ie; 2Department of Medicine, University of Cambridge, Cambridge CB2 0QQ, UK; 3Cambridge Institute of Therapeutic Immunology & Infectious Disease (CITIID), Cambridge CB2 0AW, UK; 4Cambridge University Hospitals NHS Foundation Trust, Cambridge CB2 0QQ, UK; 5Africa Health Research Institute, Durban KZN 031, South Africa; 6Hong Kong Jockey Club Global Health Institute, Hong Kong, China

**Keywords:** SARS-CoV-2, COVID-19, fusogenicity, cell-cell fusion assay, pathogenicity, infectivity, transmission, evolution

## Abstract

Early studies on the evolution of SARS-CoV-2 revealed mutations that favored host transmission of the virus and more efficient viral entry. However, cell-free virus spread is vulnerable to host-neutralizing antibodies. As population immunity developed, mutations that confer escape from neutralization were selected. Notably, cell syncytia formation wherein an infected cell fuses with a noninfected cell is a more efficient route of transmission that bypasses humoral immunity. Cell syncytia formation has been implicated in the pathogenicity of SARS-CoV-2 infection whilst compromising host transmission due to impaired whole virion release. Therefore, understanding the mechanisms of virus-mediated cell–cell fusion will aid in identifying and targeting more pathogenic strains of SARS-CoV-2. Whilst the general kinetics of cell–cell fusion have been known for decades, the specific mechanisms by which SARS-CoV-2 induces fusion are beginning to be elucidated. This is partially due to emergence of more reliable, high throughput methods of quantifying and comparing fusion efficiency in experimental models. Moreover, the ongoing inflammatory response and emerging health burden of long COVID may point to cell–cell fusion in the pathogenesis. In this review, we synthesize current understanding of SARS-CoV-2-mediated cell–cell fusion and its consequences on immune escape, viral persistence, and the innate immune response.

## 1. Introduction

SARS-CoV-2, which causes acute COVID-19 disease, continues to be a major public health concern with the emergence of mutant viral strains exhibiting enhanced ability to escape vaccine- and infection-induced antibody responses [1]. Elucidating viral entry mechanisms and SARS-CoV-2 virus lifecycle has been essential in determining the impact of emerging mutations on viral infectivity, replication, and susceptibility to vaccine-induced neutralization antibodies [2]. Similar to other Sarbecoviruses including SARS-CoV, SARS-CoV-2 establishes productive infection following successful entry and delivery of viral genomic RNA into the host cell [3]. SARS-CoV-2 enters cells through two conventional viral entry mechanisms including direct fusion with the plasma membrane and endosomal entry (Table 1). Viral entry is dependent on the binding of viral spike glycoproteins to host cell receptors on the plasma membrane, primarily angiotensin-converting enzyme 2 (ACE2) [3]. SARS-CoV-2 spike trimer comprises S1 and S2 subunits that are separated upon post-translational cleavage of the S1/S2 polybasic cleavage site by furin in infected cells [3,4]. Following engagement of ACE2 with the receptor-binding domain of the S1 subunit, cleavage of the S2′ site by endosomal or plasma membrane proteases, namely TMPRSS2, exposes the fusion peptide [5]. The fusion peptide anchors and facilitates fusion of the virion with the cell membrane [5]. The fusion peptide maintains a high mechanical stability as the heptapeptide repeat 1 (HR1) and HR2 of the S2 subunit fold back to form 6HB [5,6]. Subsequent formation of a fusion pore enables the release of the ribonucleoprotein complex into the host cell [5,6].

SARS-CoV-2 similarly induces cell–cell fusion as the spike protein expressed on infected cells interact with receptors on neighboring cells, leading to the formation of multinucleated syncytia [6]. This process shares many similarities with viral cell entry during infection—in fact, they are so similar that this phenomenon is coined as virus syncytia formation. Typically, cell fusion is a tightly regulated physiological process mediated by endogenous fusogens derived from ancestral retroviral remnants in the human genome [7,8]. For example, syncytiotrophoblast cell formation promotes immunoregulation and fetomaternal tolerance during embryological development [7,8]. However, pathophysiological cell fusion such as accidental or virus- and chemical-induced fusion promotes tetraploid cell formation and chromosomal instability associated with the development of cancer [9]. Historically coined as “fusion from within,” virus-mediated cell–cell fusion is characterized by surface expression and interaction of the spike protein on the host cell with surface receptors on neighboring cells, leading to membrane fusion and the formation of multinucleated cells [6,10]. Alternatively, “fusion from without” involves the interaction of the virus particle with adjacent cells, thereby bringing cell membranes together without viral entry (Table 1) [9].

Numerous virus families are also known to spread through cell–cell fusion via “fusion from within” including Coronaviridae, Herpesviridae, Paramyxoviridae, Retroviridae, Flaviviridae such as hepatitis C, Ebolavirus, Orthoreoviruses, and Aquareovirses [11]. Drawing together similarities of clinical outcomes between these otherwise distinct viral families provides insight into the advantages of cell–cell fusion in virus propagation. In particular, cell–cell fusion enables the transmission of virions from infected cells to adjacent uninfected cells whilst evading immune surveillance [11]. This is exemplified by SARS-CoV-2 Omicron subvariants, where cell–cell fusion is notably more resistant to mAb and serum inhibition compared to cell-free virus infection, giving rise to more efficient transmissions [12]. Cell–cell fusion leading to syncytia formation contributes to different pathological features. Syncytia formation in skin lesions is a hallmark of Herpesvirus infection [13], whilst the presence multinucleated macrophages and dendritic cells in the brain is characteristic of HIV-1 [14,15]. In SARS-CoV-2, syncytia have been observed in the lungs of COVID-19 patients [16,17].

The evolutionary trajectory of the SARS-CoV-2 spike protein contributes to viral fitness and persistence by altering infectivity, fusogenicity, and transmissibility (Table 1). Notably, the formation of multinucleated syncytia via “fusion from within” requires the completion of the fusion process which will be described in the current review. When studied through mathematical models, WT, D614G, Alpha, Beta, and Delta variants exhibited a tradeoff between the rate of syncytia formation and the speed at which the fusion process is completed [18]. Viral mutations that impact the fusion process appear to have a differential effect on the early and late stages of infection (Table 1) [18]. The fusion process is further rate-limited by cellular processes [18]. Virus evolution in favor of increased fusogenicity has been implicated in viral persistence and chronic infection in an immunocompromised patient [19]. This may play a role in determining the severity of illness and development of long COVID. Long COVID is defined as the continuation or development of new symptoms such as fatigue, dyspnea, or cognitive dysfunction 12 weeks after initial SARS-CoV-2 infection [20]. While the mechanisms underlying the heterogeneity of long COVID symptoms are unknown, cell–cell fusion may contribute to the persistence of viral RNA in the tissues and ongoing tissue damage [21]. For example, the persistence of viral fusogens was postulated to contribute to neurological symptoms in long COVID [22]. This is supported by SARS-CoV-2-induced between neurons and between neurons and glia, thereby compromising neuronal activity [22]. Moreover, long COVID is associated with sustained type I interferon (IFN) production [23]. This may be due to the activation of Cyclic GMP-AMP synthase Stimulator of Interferon Genes (cGAS-STING) system by damaged DNA in fused cells [24]. Here, we review the current research on spike-mediated cell fusion. We will also explore the possible deleterious effects of cell fusion on the innate immune response beyond immune escape and viral persistence, such as in the context of long COVID.

**Table 1 viruses-17-01405-t001:** Comparison of S-dependent viral cell entry and cell–cell fusion.

Factors	Viral Cell Entry	Cell–Cell Fusion
** Variants **	Infectivity: Omicron BA.1, BA.2, BA.5 > Delta > Omicron BA.1.1.529 > Gamma > Alpha > Beta > WT [25] BA.2.87.2~BA.2 > JN.1~WT [26]	Fusion activity: Early in infection: D614G > Alpha > Gamma > Beta > BA.1, BA.4/5 > BA.2 [27] Late in infection: Delta > D614G > Gamma > Alpha > Beta > BA.4/5 > BA.2 > BA.1 [27]
** Mutations **	Infectivity ↓: E484K mutant [28] Infectivity ↑: D614G and V367F mutant [29]	Syncytia ↓: D839Y and D839N mutant [30] T356K mutant [31] H1271K [32] Syncytia ↑: P1263L mutant [30] N354 glycosylation [31] ΔH69/V70 mutant [33]
** Cleavage **	Furin-mediated S1/S2 cleavage promotes entry but not required [34]	
** Spike location **	Surface of SARS-CoV-2 virions [30]	Surface of infected cells [30]
** Spike–receptor interaction **	Main receptor: ACE2 [30] Other receptors: CD147, AXL, KREMEN1, ASGR1 [35]	Main receptor: ACE2 [30] Other receptors: possibility of ACE2 independent entry and antibody-mediated cell–cell fusion via FcγRI [36]
** Binding Affinity **	High affinity interactions between S and ACE2 [37]	
** Energy Barriers **	High kinetic barriers due to repulsive hydration forces on phospholipid plasma membranes and viral envelope [38]	
** Tethering/Docking **	S2′ cleavage required for exposure of fusion peptide, mediated by TMPRSS2 [39] and cathepsins [40] in plasma membrane and endosomal entry routes	S2′ cleavage also required for fusion peptide exposure through TMPRSS2 mediated plasma entry pathway [39]
** NRP1 **	↑ Infection [41,42] and syncytia [22]	
** Integrins **	Do not play a role in virus syncytia formation	Syncytia ↑: integrins α_5_β_1_ [43] and α_V_β_3_ [44]

WT, wild-type; S, spike; ACE2, angiotensin-converting enzyme 2; NRP1, neuropilin-1; ↓, decreased; ↑, increased.

## 2. Multinucleated Cells in COVID-19

Early reports of COVID-19 lung pathology demonstrate diffuse alveolar damage and microvascular thrombosis underlying acute lung injury and ARDS [45]. Notably, the persistence of SARS-CoV-2 in lung pneumocytes and endothelial cells in the absence of infection of other organs led to cell syncytia formation and massive lung alterations [16]. In an analysis of post-mortem samples from 41 patients who died from COVID-19, SARS-CoV-2-infected pneumocytes with persistent viral RNA exhibited SARS-CoV-2 spike expression [16]. Infected pneumocytes displayed a dysmorphic syncytial phenotype characterized by expanded cytoplasm and inclusion bodies [16]. In another histological study of 24 deceased patients with SARS-CoV-2-related diffuse alveolar damage, 18 exhibited virus-induced multinucleated syncytia formation, which was not common in patients with non-COVID-related lung pathology [46]. Moreover, immunofluorescence and electron microscopy studies of the bronchoalveolar fluid from COVID-19 patients revealed syncytia formation between type II pneumocytes along with incomplete heterotypic neutrophil–monocyte fusion associated with disease severity [17]. Lack of syncytial cells in mild disease was attributed to a paucity of inflammatory immune activation such as infiltration of macrophages and granulocytes [17]. This would otherwise confer the need for immune escape mechanisms to safeguard viral replication and dissemination in moderate-to-severe diseases as discussed later [17].

Although multinucleated syncytia formation in other organs has not been demonstrated to our knowledge, SARS-CoV-2 has been shown to promote cell–cell fusion in other cell types in vitro (Table 2). Transfection of human-induced pluripotent stem cell-derived cardiomyocytes with SARS-CoV-2 spike protein led to multinucleated syncytia formation [47]. Cardiomyocyte syncytia demonstrated increased cellular capacitance or cell size with irregular arrhythmic activity that varied between syncytia and frequent delayed afterdepolarizations [47]. This was consistent with “calcium tsunamis” that spread along the syncytia and were greater in amplitude and duration than normal calcium transients [47]. Increased Ca^2+^ has also been observed within bridges of fused neurons and were associated with impaired neuronal activity [22]. Mouse and human brain organoids also exhibited SARS-CoV-2-mediated neuron–neuron, neuron–glia, and glia–glia fusion [22]. Therefore, cell–cell fusion may contribute to ongoing multiorgan abnormalities such as neurological symptoms in long COVID [22,48].

## 3. Evolutionary History of SAR-CoV-2: Altering the Balance Between Fusogenicity and Transmissibility

Early SARS-CoV-2 variants displayed limited fusogenic potential and developed mutations in favor of enhancing transmission. Furin cleavage was initially identified as a key step in viral entry in epithelial cell lines including Calu-3 human lung cells and primary human airway epithelial cells [49]. Meanwhile, loss of the furin cleavage site (FCS) led to a reliance on endosomal entry mediated by cathepsins, which is vulnerable to antiviral IFITM [49]. SARS-CoV-2 virus lacking the FCS demonstrated lower virus shedding in inoculated ferrets consistent with a low frequency of FCS mutations occurring in human tissues [49]. However, emergence of the more efficiently cleaved B.1.617.2 (Kappa) and B.1.1.7 (Alpha) variants bearing the P681 mutations demonstrated increased fusogenicity accompanied by compromised sensitivity to neutralizing antibodies [50].

As antibody-mediated immunity began to develop, intra-host evolution of the virus favored antibody evasion mechanisms. Mutations including N429K and Y453F at the receptor-binding motif improved ACE2 receptor binding and conferred antibody escape but at the cost of an infectivity defect [33,51]. This was compensated by the reemergence of ΔH69/V70 spike mutation [33]. ΔH69/V70 was associated with infectivity and faster fusion kinetics of the B.1.1.7 Alpha variant [33]. Similarly, the later Omicron variant bearing P681H along with N679K and H655Y mutations demonstrated escape from vaccine-elicited neutralizing antibodies [52]. Enhanced immune evasion of Omicron occurred at the expense of infectivity and cell-to-cell spread compared to Delta infection [52]. As Omicron spike is less efficiently cleaved by TMPRSS2, Omicron infection preferentially relies on entry through the endosomal route [52]. This was evident as Omicron entry was not as affected by TMPRSS2 availability compared to the Delta variant, even in the absence of ACE2 [52]. Despite initial impaired fusogenicity, Omicron variants later appeared to regain fusogenic potential as they became the dominant subvariants [53].

Further analysis of the intra-host evolution of SARS-CoV-2 during chronic infection revealed escape mutations that severely impair cell-to-cell spread [54]. P812S was found to co-occur at high frequencies with Δ69/70 and D796H mutations during late infection [54]. This was attributed to selective pressures with the administration of convalescent plasma or remdesivir treatment [54]. Upper airway specific evolution of SARS-CoV-2 demonstrated a P812S mutation in the fusion peptide that reduced S1/S2 cleavage efficiency [54]. P812S enhanced escape from serum neutralization of the WT variant containing Δ69/70 + D796H mutations [54]. However, P812S mutation severely impaired cell–cell fusion, even on the backbone of a highly fusogenic Delta spike [54].

Interestingly, mutations in other SARS-CoV-2 viral proteins have been implicated in affecting transmission and infectivity. For example, evolve-and-resequencing experiments showed that the T492I mutation in NSP4, which accelerates viral replication, increased the propensity for adaptive mutations to develop [55]. T492I accelerated the evolution of SARS-CoV-2 towards Omicron in Calu-3 cells in vitro, independent of endogenous selective pressures [55]. Taken together, SARS-CoV-2 evolution is multifaceted and continues to shape the balance between the immunogenicity, infectivity, and fusogenicity of evolving variants. However, an understanding of the key mechanisms and techniques to analyze cell–cell fusion is critical in evaluating the pathogenicity of future variants of concern, as discussed in the current review.

## 4. Mechanisms and Kinetics of Fusion

Membrane fusion is integral to a variety of processes in health and disease. Fusogens are essential for fusion as they lower the energy requirements necessary for fusion to occur. Physiological membrane fusion is exemplified by exocytosis, a key component of intracellular trafficking and secretion [56]. The process of lipid bilayer fusion is highly conserved. It is initiated by the loose tethering of two membranes by tethering factors. Fusogens, such as SNARE proteins, then strengthen this interaction by drawing membranes into close proximity. They achieve this by lowering high-energy barriers formed by repulsive membrane charges and membrane curvature [56]. The fusion of the closest monolayers gives rise to a hemi-fusion intermediate before the fusion pore opens to release vesicular cargo into the acceptor cell [56]. This process closely resembles viral cell entry, which can be mediated by three classes of fusogens that are expressed independent of the viral envelope [11,57]. SARS-CoV-2′s spike (S) protein is a class I fusogen, with a characteristic trimeric structure and high alpha-helix composition [11]. Binding of S to host receptors such as ACE2 coupled with pH changes [58] initiates the formation of the fusion pore, enabling the release of the viral ribonucleoprotein complex into the host cell (Table 1) [59]. In contrast, cell–cell fusion involves S-mediated tethering of infected cells to uninfected neighboring cells. Subsequent attachment and fusogen cleavage enable energy barriers to be overcome. These mechanisms are discussed in more detail below.

### 4.1. Cell Surface Expression of Spike

Virus-mediated cell-to-cell attachment requires cell surface expression of spike glycoproteins. Highly fusogenic viruses including paramyxoviruses, herpesvirus, and some retroviruses readily promote cell fusion as their fusogens are incorporated into the plasma membrane before budding and release [11,60,61]. However, viruses that bud from intracellular membranes including coronaviruses must bypass the localization of viral fusogens to the ERGIC during intracellular assembly [62,63]; direct trafficking of fusogens to the cell surface favors syncytia formation [62,63].

Following synthesis and post-translational modifications of the S protein, the S protein is trafficked to the ERGIC where viral particles are assembled [64]. Intracellular localization and trafficking of S depends on post-translational modifications such as S-palmitoylation and the binding of intracellular transport proteins to the cytoplasmic tail of S, as reviewed previously (Figure 1) [64]. Mapping of the S protein tail revealed a cysteine-rich proximal half embedded in the surface bilayer and a cysteine-poor distal half projecting into the cytoplasm (Figure 1) [32]. All protein interactors bind the distal portion except for the sorting nexin CNX27 which binds to the cysteine-rich portion [32]. Binding partners that interact with the cytoplasmic tail of SARS-CoV-2 S include vesicle coat proteins such as coatomer complex (COP)I and COPII that are involved in retrograde and anterograde transport, respectively [32]. COPII binds the acidic stretch DEDDSE comprising the di-acidic ER exit motif while the β’-COP subunit of the COPI complex typically binds to canonical KXKXX-type motifs [32]. The COPII complex exhibited reduced binding to the cytoplasmic tail of S containing D1257A, E1258A, D1259A, and D1260A mutations in the acidic COPII-binding motif [32,64]. This led to intracellular accumulation and impaired exit of S from the ER [32,64]. Notably, COPI binds suboptimally to the noncanonical KLHYT ER retrieval motif of the SARS-CoV-2 S cytoplasmic tail [32]. The introduction of an H1271K mutation increased COPI binding, thereby reducing S1/S2 cleavage, cell surface expression, and syncytia formation [32]. Other substitutions that have been shown to promote the accumulation of S in the ER by increasing COPI binding include the H1273A and T1273A substitutions [32]. Furthermore, T1273E and T1273D substitutions in S increase binding affinity of the β-COPI subunit by 17–fold [65]. The role of COPI binding in intracellular retention of S protein is demonstrated by an increase in the surface expression of S in the presence of a COPI sorting inhibitor [66]. Less surface trafficking of BA.1 S than B.1.1.7 S in the presence of a COPI sorting inhibitor indicates tighter COPI binding and ER retention of BA.1 S compared to B.1.1.7 S [66]. Furthermore, SARS-CoV-2 differs from other coronaviruses by lacking a tyrosine-containing motif that resembles the classic Yxxφ signal required for efficient endocytosis of surface S protein [32]. This promotes sequestration of S proteins at the plasma membrane and subsequent multinucleated syncytia formation [32]. Other binding partners including SNX27 and ERM proteins have been implicated in modifying the cellular distribution of S [64]. However, they may play a minimal role in S surface expression, since the intracellular distribution or accumulation of S on the plasma membrane was not altered by mutations in the SNX27- or moesin-binding sites [32].

The retention of S protein in the ERGIC relies on its interaction with E and M proteins [67]. For example, VeroE6 cells co-expressing S with either SARS-CoV-2 E or M but not N protein led to a predominant localization of S in the ERGIC or cis-Golgi [67]. Meanwhile, S expression on the cell surface was absent [67]. However, transfection with SARS-CoV-2 S protein alone promoted syncytia formation due to the presence of S on the cell surface [67]. This was attributed to altered S processing and maturation. The S protein exhibits increased glycosylation and poorer cleavage when co-expressed with E or M proteins, thereby reducing trafficking to the cell surface [67]. Furthermore, SARS-CoV-2 E protein was involved in reducing the kinetics and secretion of S protein, thereby further promoting intracellular retention of S [67]. Meanwhile, M protein-mediated intracellular retention of S was due to its interaction with the dibasic retrieval signaling KxHxx at the C-terminus of S. This was lost with the deletion of the last 19 amino acids of S at the C-terminus [67]. The above is consistent with the high fusogenicity of a dominant intra-host SARS-CoV-2 variant exhibiting a C25324A (S:C1254*) truncating nonsense mutation that causes a 20-amino-acid deletion at the C-terminus, reducing endoplasmic retention of S [68].

Pseudoviruses or direct transfection of spike-expressing plasmids became a ubiquitous model due to their reduced pathogenicity and, thus, reduced requirements in biosafety precautions. However, in such models where S is expressed without other structural proteins, it may not reflect accurate fusion activity as compared to live viral infections. The use of virus-like particles which contain all of S, E, M, and N may represent a more realistic representation of virus–cell membrane interaction [31,69]. In addition, the lack of active replication in a pseudovirus model limits the activation of innate immunity, adding another layer of complexity to the induced fusion activity as fusion-inhibiting restriction factors are expressed to a greater extent in the presence of active viral replication in live virus systems. The innate immune response has driven a significant evolution of accessory proteins, showing the significant role that the innate immunity plays [70].

### 4.2. Post-Translational Modifications of the Spike Protein

Both SARS-CoV-1 and SARS-CoV-2 S proteins drive cell fusion by binding and tethering infected cells to ACE2-expressing cells [39]. SARS-CoV-2 S protein is a homotrimeric class I fusion protein that must be primed by post-translational modifications to become fusion-competent [6]. Following cleavage by furin-like protease to produce S1 receptor-binding and S2 fusion fragments, S protein binds and tethers to the ACE2 receptor on the target cell membrane [71]. As S1 is shed, cleavage of S2′ by cathepsin or TMPRSS2 is a key step in the fusion event which exposes the fusion peptide for stable membrane docking and fusion (Figure 2) [71]. Notably, the capacity of SARS-CoV-2 S to be cleaved at S1/S2 and S2′ determines its relative infectivity and fusogenicity [52]. Impaired S1/S2 cleavage of the SARS-CoV-2 mutant was associated with smaller, more numerous syncytia compared to the large syncytia induced by WT SARS-CoV-2 [39]. Suboptimal cleavage of Omicron S was consistent with reduced entry efficiency and lower replication in primary 3D lower-airway organoids [52]. Omicron S also exhibited impaired cell–cell spread and fusion between spike-bearing HEK293T and ACE2-Vero cells (Table 2) [52]. Enhanced fusion efficiency and cell–cell infection observed in Delta variants B.1.617.1 and B.1.617.2 were attributed to P681R mutation in the polybasic cleavage site [50]. Meanwhile, ΔH69/V70 deletion, which enhances spike cleavage efficiency and independently emerges in multiple lineages, confers more rapid fusion kinetics in cells infected with the B.1.1.7 Alpha variant compared to WT [33]. Furthermore, using an A549-hACE2-TMPRSS2 cell-based GFP-split fusion assay, a recent study analyzed the phenotypic traits including the cell–cell fusion kinetics of recombinant viruses with 27 different spike proteins from variants of concern [27]. Consistent with the above, the Delta variant was more fusogenic than ancestral B.1, which Omicron, along with its immediate descendants, were weakly fusogenic [27]. Notably, more recently emerging Omicron subvariants including BQ.1.1, CK.2.1, CH.1.1, XBB.1, XBB.1.5, and XBB.1.16.1 exhibit higher fusogenicity than earlier Omicron spikes [27]. In fact, post-Omicron spikes, EG.5.1 and BA.2.86, displayed comparable fusogenicity to B.1 [27]. Therefore, spike mutations significantly impact the fusogenic potential of SARS-CoV-2 variants and can be studied to predict the replicative fitness, pathogenicity, and immune escape of future variants.

Post-translational modifications and conformational changes can alter the cleavage efficiency and stability of S protein and, hence, the capacity to promote cell–cell fusion. For example, the N354 glycan allosterically promotes tight adherence of the NTD and the RBD [31]. While S1 shedding is a prerequisite for cell–cell fusion, loss of the N354 glycan compromises the stability of the S trimer, thereby impairing S1 shedding and reducing cell–cell fusion activity of BA.2.86 [31]. Conversely, a single mutation S621P in BA.2.86 enhanced cell–cell fusion while the reversion P621S in XBB.1.5 markedly impaired fusion efficiency [31]. Therefore, it is thought that fusogenicity is altered by post-translational modifications and mutations that impact the stability, cleavage efficiency, and transition of S protein into the post-fusion state. Of note, other proteases including matrix metalloproteins have also been shown to promote cleavage and membrane fusion independent of the S1/S2 and S2′ site [39]. While syncytium formation may not strictly depend on furin- or TMPRSS2-mediated cleavage [39], cleavage at the S1/S2 and S2′ sites is a critical step and may predict the fusogenic potential of emerging variants.

**Table 2 viruses-17-01405-t002:** Comparison of ex vivo, in vivo, organoid, and in vitro models of fusogenicity.

Model		Method	Fusion Mediator(s)	Result	References
** Ex vivo **					
Human bronchial cartilage chondrocytes and para-bronchial gland epithelial cells		Post-mortem histological analysis	ACE2^+^ and TMEM16F^+^ staining	Syncytia found in 67% of patients (*n* = 27)	[72]
Human lung tissue pneumocytes		Post-mortem histological analysis	Not reported	Syncytia found in 87% of patients (*n* = 41)	[16]
** In vivo **					
Mice hippocampal and cortical neurons		Fluorescence fusion assay	ACE2 > TMPRSS2, NRP1	Neural fusion in up to 15% of hippocampal neurons	[22]
* C. elegans * mechanosensory neurons		Split-GFP and p15 under control of mec-4 promoter	ACE2	Fusogens observed in head, mid-body and tail being the most prevalent in ALN neurons	
Bronchoalveolar fluids of COVID-19 patients		PAP cytosmear, immunofluorescence, scanning electron microscopy, and transmission electron microscopy of bronchoalveolar fluid	Not reported	Syncytia formation between type II pneumocytes, and between monocytes and neutrophils in moderately infected patients	[17]
** Organoids **					
Human-derived brain organoids: neurons		Fluorescence fusion assay	ACE2 > TMPRSS2, NRP1	Neuronal fusion between soma and neurites	[22]
Human-derived brain organoids		Fluorescence fusion assay	ACE2 > TMPRSS2, NRP1	Neuronal fusion between soma and neurites Complete loss of neural activity	
Neurons	Glia				
Glia	Glia	Fluorescence fusion assay	ACE2 > TMPRSS2, NRP1	Neuronal fusion between soma and neurites	
** In vitro **					
HEK293T-ACE2-TMPRSS2	CHO-S	Radiometric Ca^2+^ probe	TMEM16	Ca^2+^ ↑	[73]
HEK293T expressing GFP1-10	VeroE6 expressing GFP11	Split-GFP assay	ACE2 and TMPRSS2	Fusion: Delta S > WT S; poor fusion for Omicron S	[52]
α- and ω-expressing HEK293T		Alpha complementation assay	ACE2 and TMPRSS2	SΔ19- and S-expressing virus-like particles induced fusion quantified by luminescence reaction	[74]
S-transfected HEK293T	HEK293T transfected with hACE2, CMV-Tat and HIV-LTR-FFLUC	Split lentiviral cassette (transient)	ACE2	Cell fusion detected on luminometry	[75]
S-expressing TZM-bl cell line	hACE2-expressing HOS-3734 or -3742 cells expressing hACE2	Split lentiviral cassette (stable cell line)	ACE2	Cell fusion detected on luminometry	[75]
HEK293T expressing Jun-S	HEK293T expressing Fos-ACE2 and Renilla Luciferase	BiMuC complementation	ACE2	Cell fusion detected by fluorescence measurement and altered by small molecules	[76]
HEK293T expressing VP16 activation domain	HEK293T expressing Gal4 DNA-binding domain	VP16-Gal4 transcription factor assay	ACE2 and TMPRSS2	Cell fusion impacted by mutations in S1/S2 or S2′ proteolytic cleavage; metalloproteases promote cell fusion	[39]
HEK293T-ACE2		Split-GFP assay	ACE2 and TMPRSS2	Large syncytia	[77]
A549-ACE2		Split-GFP assay	ACE2 and TMPRSS2	Large syncytia	
Vero E6		Split-GFP assay	ACE2 and TMPRSS2	No fusion upon infection but fuse when encountering Spike-expressing cells	
Caco2		Split-GFP assay	ACE2 and TMPRSS2	No fusion upon infection	
U2OS-ACE2		Split-GFP assay	ACE2 and TMPRSS2	Large syncytia	
Vero E6		Nucleus counting	ACE2 and TMPRSS2	Syncytia	[78]
Vero E6, split into two populations containing two non-functional fragments of luciferase		NanoBiT complementation assay	ACE2	Cell fusion detected on luminescence; fusion promotes TP53 stabilization	[79]
Human-induced pluripotent stem cell-derived cardiomyocytes		Immunofluorescence microscopy	ACE2; cathepsin > TMPRSS2	Syncytia; Ca^2+^ tsunamis	[47]
HEK293T		Split-GFP assay	ACE2	Syncytia; activation of cGAS-STING-IFN pathway	[24]
HEK293T		Cellular electrical impedance assay	ACE2 and TMPRSS2	Cell fusion correlated with cell index value and impacted by protease inhibitors	[80]
Huh-7 undergoing 20 serial passages		Split-GFP and Sanger sequencing	Furin cleavage site	A372T, E484D and Q493R substitution mutations arise increasing syncytia formation	[81]
Human Umbilical Vein Endothelial Cells		Immunofluorescence microscopy	Myosin light chain kinase	Syncytia; Loss of VE-cadherin at adherens junctions	[82]

S, spike; ACE2; angiotensin-converting enzyme 2; TMPRSS2, transmembrane serine protease 2; NRP1; neuropilin-1; GFP, green fluorescent protein; CMV-Tat; cytomegalovirus promoter-driven Tat; HIV-LTR-FFLUC; HIV long terminal repeat driving firefly luciferase; BiMuC, bimolecular multicellular; ↑, increased.

### 4.3. Tethering, Docking and Fusion: Overcoming Energy Barriers

Following spike tethering to a neighboring cell, high-energy barriers must be overcome for membrane fusion to occur. Membranes that span a small radius such as small vesicles require little energy to form the fusion pore [83]. However, the thermodynamic and biophysical requirements are markedly higher for cell–cell fusion. Cell fusogens initiate tight adhesion at approximately 10nm of membrane separation [84]. Once adhesion is established, four main energetic barriers must be overcome to initiate cell fusion and are associated with three hallmarks of cell–cell fusion [84]. First is the dehydration of contacting plasma membranes [84]. Ca^2+^-dependent membrane ordering and dehydration contributed to more efficient viral entry of SARS-CoV-2 compared to SARS-CoV-1 [85]. SARS-CoV-2 fusion peptide exhibited greater cooperativity of the Ca^2+^ binding sites and hydrophobicity [85]. Calcium facilitates fusion of the SARS-CoV-1 envelope with the cell membrane by shielding repulsive negative charges on the fusion peptide and promoting membrane insertion, thereby allowing viral cell entry [86]. Moreover, calcium complexes with and promotes formation of anionic lipid clusters that induce a spontaneous negative curvature known to facilitate fusion [87]. Although the role of Ca^2+^ binding in SARS-CoV-2-mediated cell–cell fusion has not been explored to our knowledge, Ca^2+^ has been reported to contribute to syncytia formation in viral infections. For example, BSC-1 cells expressing the HIV-1-enveloped glycoprotein (gp120-gp41) were previously shown to bind and form syncytia with CD4-expessing HeLa cells, which was markedly reduced in the absence of calcium ions in solution [88]. Therefore, extracellular Ca^2+^ likely facilitates spike-mediated cell fusion by reducing biochemical energy barriers such as repulsive negative charges that limit initial plasma membrane contact.

After membrane dehydration and overcoming repulsive charges, formation of the hemi-fusion intermediate and pore widening relies on cholesterol-specific interaction [89]. Cholesterol is suggested to stabilize the pre-pore structure while facilitating lipid mixing and fusion pore expansion independent of lipid rafts [46,89]. Cholesterol was previously shown to interact with the transmembrane domain of influenza spike protein, hemagglutinin [89]. Cholesterol promoted faster lipid mixing from RBCs to Sf9 insect cells stably expressing hemagglutinin prior to the fusion pore opening [89]. The presence of cholesterol allowed fusion to go to completion and promoted pore expansion which was stunted in the absence of cholesterol [89]. Similarly, drugs that interfere with cell membrane cholesterol such as 25-hydroxycholesterol or methyl-beta-cyclodextrin reduce SARS-CoV-2-induced syncytia formation in a dose-dependent manner [6,46,90,91]. Moreover, externalization of phosphatidylserine further potentiates SARS-CoV-2-mediated membrane fusion [73]. Phosphatidylserine externalization is typically associated with platelet activation and initiation of coagulation [92] but is also a conserved signal for membrane fusion [93]. TMEM16F lipid scramblase, ANO6 promoted phosphatidylserine scrambling which was dependent on SARS-CoV-2 S interaction with ACE2 [73]. Inhibition of ANO6 ion channel markedly reduced SARS-CoV-2-induced phospholipid scramblase activity and multinucleated cell formation [73]. Notably, ANO6 induced an increase in cytosolic Ca^2+^ that facilitated lipid mixing and hemi-fusion after initial cell–cell adhesion leading to syncytia formation between HEK293T-ACE2-TMPRSS2 and CHO-S cells infected with SARS-CoV-2 pseudovirus [73]. Therefore, SARS-CoV-2 influences cell–cell fusion kinetics in favor of overcoming energy barriers that are known to limit formation and expansion of the fusion pore.

## 5. Viral Factors That Influence Cell–Cell Fusion

### 5.1. ACE2-Dependent Fusion

Tethering of SARS-CoV-2 conventionally relies on the interaction between the SARS-CoV-2 S protein and ACE2 expressed on the surface of infected and target cells, respectively. Studying the binding affinity of the S protein for the ACE2 receptor has aided in understanding viral infectivity of emerging variants, but its impact on fusogenicity is still being explored. Interestingly, S mutations, L48S and A372T, in the RBD was associated with increased ACE2 interaction and markedly increased cell–cell fusion compared to Wuhan-Hu-1 S by approximately 6- and 17-fold, respectively [81]. These mutations conferred a replicative fitness advantage and increased syncytia formation under low-ACE2 conditions [81]. This suggests that high affinity interactions may be critical in overcoming high-energy barriers even at low frequency. ACE2 localization on the cell surface such as on lipid rafts also has a minimum effect on cell–cell fusion [94].

S interaction with ACE2 can be further enhanced by cofactors such as integrins (Table 2). The role of integrins in SARS-CoV-2-mediated viral entry remains controversial. Activated integrin α_5_β_1_ has been reported to facilitate viral entry in cells not expressing ACE2 via endocytosis but not virus syncytia formation [95]. Another study showed that α_5_β_1_ directly interacts with the S protein independent of the Arg–Gly–Asp motif [43]. α_5_β_1_ markedly enhanced fusion but not infection of ACE2-expressing HEK293T cells compared to α_5_β_1_ integrin truncated at the cytoplasmic tail of α_5_ [43]. Discrepancies in the mechanisms of viral cell entry and cell–cell fusion may be due to differences in the energy requirements needed to overcome fusion barriers; certain high-affinity interactions may only confer a fusion advantage in the presence of higher-energy barriers.

### 5.2. ACE2-Independent Fusion

Interestingly, cell–cell fusion mechanisms are not limited to ACE2-dependent interactions. S protein has been reported to interact with cell receptors independent of ACE2 which may contribute to aberrant syncytia formation amongst cell types typically expressing low levels of ACE2. Moderate-to-severe COVID-19 is associated with the formation of heterotypic syncytial cells comprising infected neutrophils and monocytes [17]. Although ACE2 has been implicated in infection and activation of neutrophils and macrophages [96,97], only a small proportion of monocyte-derived macrophages expressed ACE2 around the sites of SARS-CoV-2 infection, suggesting an ACE2-independent mechanism of infection and cell–cell fusion [98].

Antibody-mediated cell–cell fusion independent of ACE2 has been described [99]. Receptor-binding motif (RBM)-specific neutralizing antibodies, CB6 and RGN10933, more robustly induced the fusion of ACE2-deficient HEK293T cells expressing WT spike compared to non-RBM targeting antibodies [99]. Antibody-mediated cell–cell fusion was attributed to proteolytic cleavage of the S1/S2 site and shedding of S1 [99]. Furin inhibition abrogated CB6-induced activation of WT spike and restored the capacity of CB6 to neutralize cell–cell fusion [99]. CB6 also failed to induce fusion of HEK293T cells expressing the R685A spike mutant which lacks the S1/S2 furin cleavage site [99]. Meanwhile, CB6-induced fusion was restored by the shedding of S1 with the addition of extracellular trypsin [99]. Antibody-mediated spike activation was further shown to promote cell-to-cell transmission of authentic virus infection [99]. CB6 induced a concentration-dependent increase in S2′ cleavage and N protein in SARS-CoV-2 infected VeroE6 cells independent of ACE2 [99]. While antibodies targeting the RBM may contribute to ACE2-independent cell fusion, it is unclear if fusion-promoting antibodies such as CB6 contribute to antibody-dependent enhancement (ADE).

Another study observed fusion between S-expressing HEK293T cells and FcγRI-expressing cells decorated with monoclonal antibodies or antibodies from convalescent patient serum samples [36]. Cell–cell fusion was markedly induced by anti-RBD antibodies C63C8, G32B6, and S2H97 but not by antibodies directed against the NTD [36]. This was recapitulated in cells expressing mAb-ACE2T chimeras in which the RBD-binding site of ACE2 was replaced with the antigen-binding fragment of the IgG antibody [36]. However, impaired binding to BA.2 Omicron variant diminished fusion activity [36]. For example, the interaction of C63C8-ACE2t-expressing cells with BA.2 S demonstrated 17% of the fusion activity induced by ACE2, consistent with weaker affinity of C63C8 for BA.2 S than for WT S [36]. Interestingly, SP1-77-ACE2t and C81D6-ACE2t did not induce cell fusion despite potently neutralizing and binding the Omicron variant, respectively [36]. C63C8-ACE2t, G32B6-ACE2t, C12A2-ACE2t, and S2H97-ACE2t chimeric constructs were also shown to promote the infection of authentic SARS-CoV-2 virus in MDCK cells [36]. This highlights the potential for fusion-promoting antibodies to also contribute to ADE.

The risk of ADE was a major concern for vaccine development early in the pandemic and has been reviewed previously [100]. A subset of monoclonal antibodies have been observed to induce ADE of SARS-CoV-2 infection in vitro [101]. mAbs specific to SARS-CoV-2 spike were isolated by antigen-specific B cell sorting from donors previously exposed to SARS-CoV-2 and were characterized [101,102]. Anti-RBD IgG1 mAbs were shown to enhance THP-1 infection comparable to or even exceeding levels observed in permissive HeLa-ACE2 cells [101]. High levels of infection were similarly recapitulated in monocyte-derived macrophages by several infection-promoting mAbs [101]. Infection by mAbs was further enhanced in the presence of ruxolitinib, a JAK1/2 inhibitor [101]. Specific monoclonal antibodies found in convalescent sera promote infection but fail to overcome neutralization by polyclonal sera [101]. It is thought that the deleterious effects of infection-promoting antibodies would be masked by the complex polyclonal antibody response [101]. However, an increased potential for ADE may be observed under a suboptimal polyclonal antibody response during peak viral replication [101]. Whether ADE enhances the clinical severity of the illness remains unknown [100].

Importantly, macrophage infection secondary to ADE was associated with the production of proinflammatory cytokines including CXCL9, IP10, IL-19, IFNα, and TNFα, and notably, the formation of multinucleated syncytia [101]. Given the antibody-dependent enhancement of SARS-CoV-2 infection [101], further studies are needed to test if infection-promoting mAbs also play a role in antibody-mediated cell–cell fusion of FcR-expressing cells. This may reveal an important relationship between antibody-mediated infection, pathogenicity of SARS-CoV-2, and ongoing inflammation associated with persistent symptoms.

Dual cleavage of S protein has been implicated in ACE2-independent cell–cell fusion in HEK293T cells [103]. This was dependent on cleavage by furin, but not TMPRSS2 [103]. Unlike other viral fusogens such as HIV Env, CMV gB, Influenza HS, and SARS-CoV-1 S, SARS-CoV-2 S is cleaved twice during its biosynthesis pathway [103]. Primary cleavage in the trans-Golgi network followed by additional cleavage by furin in trans was suggested to be sufficient for the initiation of the fusion process [103]. Meanwhile, S proteins that are inefficiently cleaved rely on ACE2 binding [103]. This is consistent with the inability of Omicron to induce cell–cell fusion in the absence of ACE2 due to inefficient S cleavage by furin [103]. Others have also reported the regulation of cell–cell fusion activity by S1/S2 cleavage independent of ACE2 affinity [12]. Despite exhibiting a higher binding affinity to ACE2, BA.1 showed reduced cell–cell fusion activity compared to D614G spike due to inefficient S1/S2 processing [12]. Although furin is thought to serve as an entry mediator in the initiation of cell–cell fusion by efficiently cleaving S [103], binding of SARS-CoV-2 S to other membrane proteins have been reported to contribute to virus infection [104], but their role in cell–cell fusion remains to be explored.

## 6. Host Factors That Influence Cell–Cell Fusion

Different cell types vary in fusogenicity following SARS-CoV-2 infection [77]. Cells were infected with SARS-CoV-2 and examined for the formation of syncytia in five cell lines [77]. Evidence of cell–cell fusion was found in U2OS, A459, and HEK293T cell lines but not Caco2. Vero E6 cells were unique in that they did not fuse upon direct infection but demonstrated fusogenic properties upon encountering infected cells or cells expressing S. Endogenous ACE2 expression was, therefore, sufficient to trigger fusion [77]. Fusogenic properties seen in different cell lines have also been observed in post-mortem analyses of COVID-19 patients, with reports of syncytia formation exhibited by pneumocytes in the lungs [105]; in neurons and glia in the brain [22]; and in intestinal epithelial cells in the gastrointestinal tract [106]. However, differences in fusogenic properties in different cell types have not been studied in vivo. In this section, we will discuss the different host factors that could contribute to differences in fusogenic properties observed in cell types.

### 6.1. Impact of IFITMs on Cell–Cell Fusion

Interferon-induced transmembrane proteins (IFITMs) exhibit an inhibitory effect on cell–cell fusion as juxtaposed to TMPRSS2 [77]. IFITM production is triggered by interferon release in infected cells that results in interferon-stimulated gene expression [77]. Five types of IFITMs exist, and their relative abundance is dependent on cell type [77]. IFITM1 has the greatest effect in inhibiting cell–cell fusion in U2OS, A459, HEK293T, Caco2, and Vero cells. Meanwhile, IFITM2 and IFITM3 are ineffective in U2OS and HEK293T cells [77]. This difference in potency is likely due to the localization of IFITM in the cell: IFITM1 is found at the plasma membrane and can prevent fusion by membrane content or rigidity modification, whereas IFITM2 and IFITM3 are at the endo-lysosomal compartment [77]. Interestingly, this inhibitory effect is counteracted by TMPRSS2 activity, although the underlying mechanism is not fully understood in SARS-CoV-2. It is thought that proteolytic pathways, seen in WIV1 [107] and HCoV-299E [108], could be one way of counteracting IFITMs where cleavage at the S2′ site results in exposure of the fusion peptide, thereby initiating membrane fusion.

### 6.2. Impact of Temperature and Circadian Rhythm on Cell–Cell Fusion

Temperature of the host environment can also alter cell–cell fusion in addition to protein and lipid expression, which may vary along the respiratory tract and during the secretion of pyrogenic mediators such as IL-6, IL-8, and PGE_2_ [109]. This temperature dependent effect is observed at initial stages of attachment [110] and at S1/S2 cleavage, with each stage showing the contrasting effects of temperature [111]. Overall, lower temperatures are associated with decreased cell–cell fusion [111]. Binding affinity of S to ACE2 shows a stepwise increase at lower temperatures leading to decreased dissociation kinetics [110]. These differences translate into biological differences where increased S interaction and viral attachment are seen at lower temperatures [110]. On the other hand, decreased S1/S2 cleavage at the FCS is seen at 33 °C and 37 °C compared to 39 °C [110]. Sequencing analysis revealed that the FCS was prone to an increased accumulation of mutations at lower temperatures after 20 passages in Vero-E6 and HEK293T cells [110]. FCS mutations such as S686G were thought to provide a fitness advantage by decreasing cell fusion and promoting higher viral yields at 33 °C and 37 °C [110]. However, this fitness advantage was lost at 39 °C due to increased cell–cell fusion even with FCS impairment [110]. Pyroptotic cell death of syncytial cells is thought to play a role in the excessive inflammatory response in severe COVID-19 [112]. While whole virion release and reduced cell fusion may facilitate rapid asymptomatic spread of SARS-CoV-2, syncytia formation may drive the pathogenesis and progression to severe disease in febrile patients.

Moreover, circadian rhythm is implicated in cell–cell fusion by regulating ACE2 expression [113]. Silencing of *Bmal1*, a major circadian transcription activator, led to reduced ACE2 expression in Calu3 lung epithelial cells [113]. Bmal1 repression with REV-ERB agonism decreased ACE2 expression, thereby limiting SARS-CoV-2-induced cell–cell fusion of Huh-7 cells [113]. Therefore, a number of host factors that regulate spike protein modifications and target receptor expression are important determinants of fusion efficiency. These may, in turn, alter the clinical course of the disease.

## 7. Indirect Methods of Measuring Cell–Cell Fusion

Conventional methods of measuring cell–cell fusion involve counting multinucleated cells. However, numerous techniques have been generated to more accurately quantify virus-mediated syncytia formation (Table 2). We define indirect methods as an estimate of cell–cell fusion that relies on visualizing changes in the relative distribution of molecules as cells fuse. This is based on the principle that small cytoplasmic elements and soluble proteins including fluorophores can diffuse across cytoplasmic bridges to adjacent fused cells [22]. For example, calcium (Ca^2+^) distribution has been used to detect fusion of excitable tissues. Fused cardiomyocytes exhibit “calcium tsunamis” that are measured with the calcium indicator Fluo-4 and used to quantify percent of syncytia formation [47]. Ca^2+^ imaging also showed increased intracellular Ca^2+^ concentrations within bridges of fused neurons in human and mouse brain organoids [22].

In addition to estimating cell–cell fusion based on Ca^2+^ localization, analysis of the distribution of fluorescent signals has also been implemented. This could either be reflected by a movement in fluorescent molecules between interconnected cells, or a change in the overall morphology marked by multinucleated giant cells. Diffusion of a photoconvertible fluorescent protein Kaeda was used in a study of fusion between neurons expressing SARS-CoV-2 spike protein and human ACE2 [22]. Once green-fluorescent interconnected neurons were identified, pulses of UV light were applied to photoconvert the green Kaeda fluorophore to red Kaeda fluorophore molecules that rapidly diffused to the adjacent neuron [22]. Bidirectional organelle exchange between fused neurons was further demonstrated by tagging mitochondria with two different fluorophores, mito-mPA-GFP and mito-mPA-mCherry 1 [22].

Immunofluorescence can also be implemented to detect morphological changes associated with cell–cell fusion. In murine hippocampal cultures, fusion of GFP- and mCherry-expressing populations demonstrated giant cells expressing both GFP and mCherry [22]. Moreover, cardiac markers including α-actinin and sarcoplasmic reticulum markers such as calreticulin and SERCA2a were stained to visualize multinucleated clusters of fused human-induced pluripotent stem cell-derived cardiomyocytes transfected with SARS-CoV-2 spike protein [47].

Therefore, while indirect methods may help visualize cell–cell fusion, they are relatively nonspecific compared to direct methods. Direct methods measure the output of a biochemical reaction that takes place during cell–cell fusion. It often involves two complementary protein fragments that are non-functional in isolation but generate a signal when they interact. Specific direct methods are described below.

## 8. Direct Methods of Measuring Cell–Cell Fusion

### 8.1. Split-GFP Assay

The split-GFP assay is a robust approach to study protein–protein interactions in both in vitro and living cells [114]. This assay utilizes two different fragments of GFP that are non-fluorescent individually but generate a fluorescent signal upon physical interaction [115]. In the context of SARS-CoV-2 cell–cell fusion studies, these GFP fragments are expressed in two separate cell populations. The fusion that takes place between these cells facilitates the physical interaction of the two GFP fragments, leading to the reconstitution of fluorescence that can be visualized directly under a microscope. This method has been employed in a study conducted using Vero E6 cells expressing GFP1-10 and HEK293T cells expressing GFP11 to monitor the fusion events mediated by SARS-CoV-2 variants such as the Delta and Omicron variant [52]. Despite its high specificity, the split-GFP assay is limited by the stability of fluorescence and quantification over time, requiring precise optimization of experimental conditions [116].

### 8.2. Split Lentiviral Cassette Assay

The split lentiviral cassette assay was developed early in the COVID-19 pandemic to complement routine neutralization assays in assessing SARS-CoV-2 spike–hACE2 interaction [75]. Two systems were developed that are based upon HIV long terminal repeat (LTR) activation with HIV Tat and involve either transient co-transfection of HEK293T cells or stable cell lines [75]. In the transient system, separate populations of HEK293Ts were transfected with plasmids encoding S and hACE2 along with CMV promoter-driven Tat (CMV-Tat) and HIV LTR driving firefly luciferase (HIV-LTR-FFLUC), respectively. Upon fusion of co-transfected S-expressing cells with target hACE2 cells, Tat-mediated transactivation of HIV LTR leads to FFLUC expression detected on luminometry [117]. While the transient cell fusion system is a more robust measurement of serum inhibition than the pseudotype-based neutralization assay, it is not suitable for high throughput use due to its unreliability and complicated procedure [75]. Therefore, as they have been widely implemented in HIV pseudotyping assays, TZM-bl cells with integrated Tat-responsive HIV-LTR-FFLUC and HIV-LTR-LacZ cassettes were used to generate a stable cell line to study SARS-CoV-2 fusogenicity [75]. Fusion was measured between TZM-bls stably expressing spike and hACE2-expressing HOS-3734 or -3742 cells that were transduced with a third-generation HIV vector with an intact Tat. Although use of stable cell lines creates a highly reproducible cell fusion assay without the need for additional reagents, extracellular release or variable binding of Tat may impact transactivation and is more complex than fragment-based fusion assays such as the split-GFP assay [117,118].

### 8.3. Alpha Complementation Assay

The alpha complementation assay is based on beta-galactosidase which was previously utilized to assess HIV-mediated fusion [119]. Following cell–cell fusion, the omega (ω) enzyme fragment joins with the alpha fragment to form active beta-galactosidase, which generates a measurable chromogenic or luminescent signal [74]. This assay has been implemented in research using HEK293T and Vero E6 cells that express the spike protein of SARS-CoV-2 and ACE2 receptor to quantify fusion activity [74]. While chromogenic detection in this assay is relatively straightforward, luminescence-based readouts are favorable due to their dynamic range and sensitivity, especially in fusion scenarios with low frequencies [120].

### 8.4. Bimolecular Multicellular (BiMuC) Complementation Assay

The bimolecular multicellular (BiMuC) complementation assay was initially developed for the detection of Nipah virus-induced syncytia formation [121], and was more recently adapted for measuring the fusogenicity of SARS-CoV-2 [76]. The assay relies on the reconstitution of venous fluorescent proteins (VFP) during the interaction of two complementary partner proteins c-Jun and c-Fos fused with VFP fragments VN and VC, respectively [76]. Viral-induced fusion between separate cell populations expressing c-Jun/VN and c-Fos/VC chimeras generates a fluorescent signal. The BiMuC approach can be broadly implemented in different cell lines to study the fusogenicity of different viruses. In addition to SARS-CoV-2-induced syncytia formation in HEK293Ts, this assay effectively demonstrated influenza A virus- and Nipah-induced fusion of MDCK cells and HEK293Ts, respectively, with or without the presence of select inhibitors [76]. Importantly, the BiMuC assay was used to screen over 1000 diverse small molecules to study their impact on viral-mediated cell–cell fusion [76]. Despite its flexibility and high throughput implementation without the need for microscope-based equipment [76], the BiMuC assay may face similar limitations as the transient lentivirus cassette system that impact reliability and reproducibility.

### 8.5. NanoBiT Complementation Assay

The NanoBiT complementation assay leverages two non-functional fragments of luciferase, resulting in luminescence upon fusion [122]. The two fragments are expressed in different cell populations, and the interaction after fusion results in a strong luminescent signal [123]. This assay has been widely adopted in the study of the fusion kinetics of SARS-CoV-2 variants, particularly in Vero E6 cells [79]. The high temporal resolution of this assay allows real-time monitoring of fusion events. However, meticulous optimization of fragment expression and cell seeding densities is required to minimize background signal [124].

### 8.6. VP16-Gal4 Transcription Factor Assay

The VP16-Gal4 transcription factor assay combines transcriptional activation with cell–cell fusion detection [125]. In this assay, one cell population expresses the VP16 activation domain, while the other cell population expresses the Gal4 DNA-binding domain. Fusion between these cells reconstitutes a functional transcription factor that results in the expression of a reporter gene, such as luciferase [126]. This assay has been validated in studies that use HEK293T cells to explore the molecular mechanisms of SARS-CoV-2 fusion. For instance, this assay was used to examine the fusogenicity of SARS-CoV-1 and SARS-CoV-2, including in the presence of protease inhibitors [39]. While being quantitative and highly sensitive, VP16-Gal4 transcriptional activation may present variability across different cell types, such as in primary airway epithelial cells, highlighting the importance of maintaining standardized conditions [127].

### 8.7. Cell–Cell Electrical Impedance (CEI)

Cell–cell electrical impedance (CEI) is a label-free, powerful technique that monitors the changes that occur in electrical impedance as cells fuse [128]. This method generates kinetic data based on the disruption of the cell membrane during the fusion event, thereby avoiding the additional need for reporters or labels [129]. CEI has been successfully employed in studies involving cells such as Huh7 and HEK293T cells to monitor spike-mediated fusion events as well as the modulation by entry inhibitors [130]. A study conducted in HEK293T cells demonstrated the use of CEI in determining the impact of TMPRSS2 protease inhibitors on SARS-CoV-2 fusion kinetics [80]. Despite the non-invasive nature of CEI, its lower specificity for molecular events associated with fusion warrants the need for validation with complementation methods such as the NanoBiT and BiMuC assays [131].

## 9. Spike-Mediated Cell–Cell Fusion in Immune Escape, Virus Dissemination and Persistence, and Sustained Type I Interferon Production

### 9.1. Immune Escape and Immunosuppression

It is thought that cell-to-cell transmission evolved as an immune escape mechanism to evade neutralizing antibodies that bind free virions. Serum inhibition of cell–cell fusion between SARS-CoV-2 S-expressing HEK293Ts and hACE2-expressing TZM-bl cells (TZM-bl-hACE2) showed three-fold lower ID50 titer compared to cell-free pseudovirus neutralization data of TZM-bl-hACE2 cells [12]. Furthermore, fusogenic SARS-CoV-2 variants such as BA.2.75 and BA.4/5 were less sensitive to neutralization with monoclonal antibodies or vaccinee sera following the third dose of BNT162b2 Pfizer–BioNTech COVID-19 vaccine compared to BA.2 [132]. Interestingly, post-translational modifications including the acquisition of N354 glycosylation by K356T substitution was associated with enhanced fusogenicity and conferred escape from broadly neutralizing antibodies E1, E2.1, and E2.2 isolated from vaccinated or breakthrough infection individuals [31]. N354 glycosylation also impaired E-mediated antibody-dependent cell cytotoxicity and markedly reduced immunogencity in a model of real-world mimicry immunity background in mice [31]. Therefore, the fusogenicity of SARS-CoV-2 variants, whether independent or dependent on S neutralizing epitopes, is likely associated with immunogenicity and escape from the host response. An understanding of the mechanisms and kinetics of spike-mediate cell fusion will aid in identifying molecular targets to abrogate fusion and improve the immunogenicity against SARS-CoV-2 variants.

In addition to the humoral immune response, cell–cell fusion has also been shown to impact innate and adaptive immunity. Heterotypic cell-in-cell structures were previously found in the lung autopsies of COVID-19 patients which led to cell death of internalized lymphocytes [78]. Strikingly, the frequency of syncytia containing CD45^+^ cells was inversely associated with the number of circulating peripheral lymphocytes, indicating a role for cell–cell fusion in lymphocyte depletion [78]. Multinucleated 293T-ACE2 cells transfected with SARS-CoV-2 S glycoprotein readily internalized CCRF-CEM human T cell leukemia lymphoblasts expressing mCherry as well as other leukocyte cell lines including THP-1 monocytes, Jurkat T lymphocytes, Raji B cells, K562 human erythroleukemia cells, and PBMCs [78]. Interestingly, internalization and cell death preferentially affected CD8^+^ over CD4^+^ PBMCs [78]. The mechanisms of leukocyte internalization by multinucleated syncytia are unclear. Initially, chemoattractant release and antigen presentation would promote lymphocyte recruitment and target multinucleated syncytia for cytotoxic cell death. However, recruited lymphocytes may be internalized upon interaction with syncytial cells via kinetics potentiated by membrane phospholipid reorganization and the curvature of the syncytial cell surface [133].

Lymphocyte internalization is distinct from heterotypic syncytial cells formed by the fusion of infected neutrophils and monocytes in COVID-19 patients at the moderate stage (9–16 days) of the disease [17]. Aberrant leukocyte recruitment associated with severe disease led to the formation of matured syncytial cells exhibiting a surface morphology resembling that of a macrophage [17]. Therefore, cell–cell fusion may impair host immunity with excess fusion of recruited leukocytes in severe disease.

### 9.2. Syncytia Formation in Acute and Chronic Inflammation

Although cell–cell fusion aids in immune escape and viral replication, the advantages of cell–cell fusion for virus transmission may be limited by the cell death of multinucleated syncytia. SARS-CoV-2-mediated syncytia formation induces pyroptotic cell death via GSDME activation [112]. Fusion of HeLa-S and HeLa-ACE2 cells led to syncytia growth and rupture [112]. This led to the release of lactate dehydrogenase and increase in caspase-3/7 activity which was abrogated with a pan-caspase inhibitor [112]. Fused THP-1-ACE2 and HeLa-S cells also released IL-1β [112]. GSDME knockout did not affect syncytia formation but inhibited the death of syncytia [112]. The cytopathic effects of SARS-CoV-2-induced cell–cell fusion is further exacerbated by the stabilization of TP53 [79]. SARS-CoV-2-induced membrane fusion of Vero E6 cells was shown to induce TP53 stabilization in a pH-dependent manner [79]. TP53 stabilization was associated with fusogenic SARS-CoV-2 variants Alpha/B.1.1.7 and Delta/B.1.617.2 compared to WT [79]. TP53 regulated chromatin accessibility, expression of microRNAs, cellular senescence, and proinflammatory cytokine release [79]. The cytopathic effects of syncytia formation may drive acute inflammation in severe SARS-CoV-2 infection while potentially offsetting the advantages of cell fusion for viral transmission.

The role of syncytia formation in inflammation is further highlighted by the release of type I IFNs from infected multinucleated cells [134]. Analysis of S-mediated syncytia formation revealed transcriptional upregulation of *IFNβ*- and IFN-stimulated genes [24]. Conversely, *IFNβ* expression was reduced in cocultures of HeLa-ACE2 cells with either a knockout in *cGAS* or *STING* [24]. Therefore, type I IFN production in fused cells is cGAS-STING-dependent. cGAS-STING-induced type I IFN production is a critical component in the innate immune system and is triggered by pathogen- or host-derived DNA [135]. cGAS-STING activation relies on efficient S cleavage necessary for cell fusion to occur [24]. In fused cells, cGAS colocalizes in micronuclei and is activated by DNA aggregates following nuclear membrane rupture [24]. Subsequent downstream IRF3 activation promotes IFN production [24]. Therefore, syncytia formation may underlie the inflammatory response associated with cGAS-STING signaling, potentially impacting COVID-19 disease severity and symptoms. This is supported by elevated gene expressions of *cGAS* and *STING* in peripheral blood leukocytes and higher plasma IFN-α in severe (*n* = 44) and long COVID (*n* = 30) compared to non-severe (*n* = 43) and convalescent (*n* = 31) COVID-19 [23].

Long COVID is defined as the continuation or development of new symptoms such as fatigue, dyspnea, or cognitive dysfunction 12 weeks after initial SARS-CoV-2 infection [20] but can persist for over a year after disease onset. As the burden of long COVID increases with a widely varying prevalence from 6% to 50% [136], an understanding of the underlying pathogenic mechanisms would guide therapeutic strategies to prevent post-infection syndromes. The presence of multinucleated syncytia in the tissues of long COVID patients have not been reported to our knowledge. The persistence of SARS-CoV-2 RNA in long COVID [137] may potentiate syncytia formation, thereby driving ongoing inflammation and type I IFN production. Moreover, the passive transfer of IgG from long COVID patients with persistently elevated IFN-α and IFN-β to mice led to muscle immobility [138]. Further studies are needed to elucidate the role of cell–cell fusion and persisting type I IFN in long COVID.

## 10. Conclusions

Viruses such as SARS-CoV-2 conventionally act as obligate intracellular parasites and transmit infection following assembly and budding of whole virions from the intracellular membranes of infected cells. Release of virions into the extracellular space provides an opportunity for further cells to be infected but is susceptible to neutralization by the humoral immune response. Fusion of infected cells with neighboring cells allows direct cell-to-cell transmission of virus material known as “fusion from within”, thereby evading host immunosurveillance. The fusion process is required for syncytia formation and is impacted by both viral and host factors. Implementing robust high throughput methods to study the fusion process has been useful in monitoring the phenotypic evolution of SARS-CoV-2 and in aiding our understanding of cell fusion. However, further studies are needed to elucidate the role of syncytia formation in the pathogenesis of viral infections such as COVID-19. This will help to identify and target selective pressures that impact fusogenicity and infectivity and, therefore, the balance between the pathogenicity and transmissibility.

## Figures and Tables

**Figure 1 viruses-17-01405-f001:**
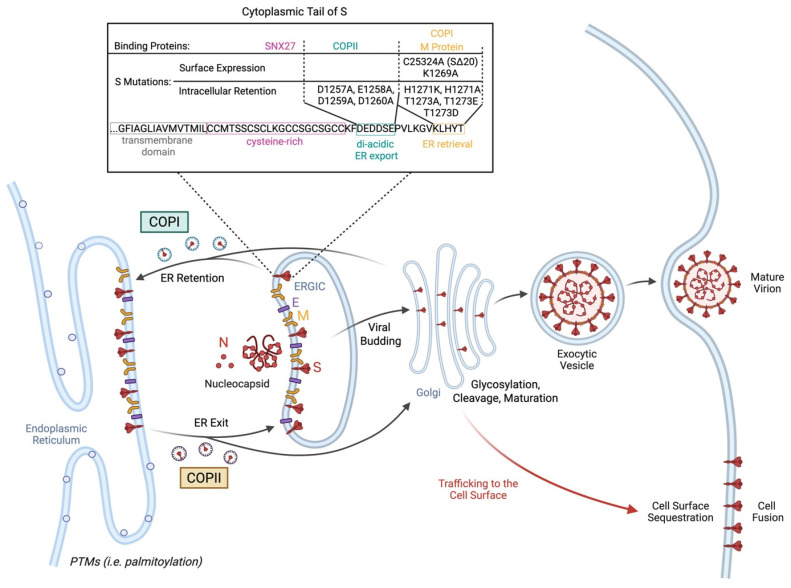
Spike protein trafficking to the cell surface. Intracellular localization of spike (S) protein in the ERGIC is necessary for viral budding and generation of the SARS-CoV-2 virion. S trafficking is highly regulated by transport proteins such as coatomer complex (COP)I and COPII that mediate retrograde and anterograde transport, respectively. Meanwhile, SARS-CoV-2 M protein binds and retains S in the ER. Mutations and post-translational modifications (PTMs) including S-palmitoylation can impact attachment of binding partners to the cytoplasmic tail of S, thereby interfering with the intracellular retention of S for virion formation. This allows S to escape to the surface and promote cell fusion. Mutations in S that reduce binding of COPII to the acidic stretch DEDDSE and, therefore, impair exit from the ER including D1257A, E1258A, D1259A, and D1260A. Meanwhile, reduced binding affinity of COPI or M protein to the cytoplasmic tail of S impairs intracellular retention and subsequently promotes sequestration of S at the plasma membrane necessary for cell–cell fusion. A C25324A missense mutation that causes a 20-amino-acid deletion at the C-terminus promotes surface expression of S protein and fusion, likely due to inability of M protein to bind and anchor S in the ERGIC for viral budding. Meanwhile, suboptimal binding of COPI to the noncanonical KLHYT ER retrieval motif further promotes trafficking of S to the cell surface. Impaired COPI binding is exacerbated by the K1269A mutation leading to increased syncytia formation. However, increased binding affinity of COPI to the cytoplasmic tail of S carrying the H1271K, H1271A, T1723A, T1273E, or T1273D substitutions promotes intracellular retention of S in the ER, thereby reducing surface expression and syncytia formation. Created with https://BioRender.com.

**Figure 2 viruses-17-01405-f002:**
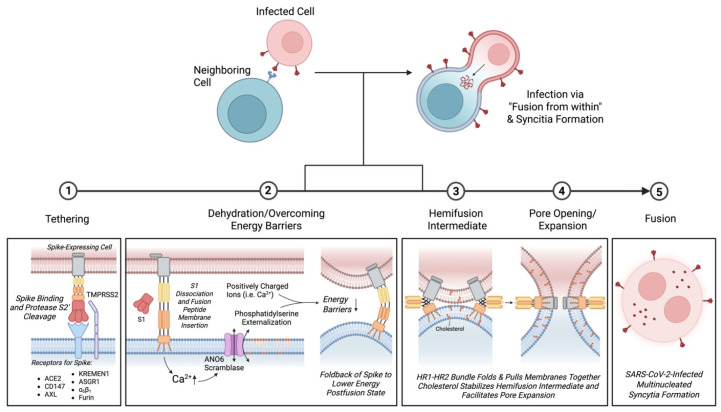
Kinetics of SARS-CoV-2 multinucleated syncytia formation. Fusion of a SARS-CoV-2-infected cell with a neighboring cell promotes viral transmission known as “fusion from within.” **Tethering**: Syncytia formation is initiated by tethering of a spike-expressing cell to host receptors on the surface of an uninfected cell. While ACE2 is the primary receptor for the SARS-CoV-2 spike protein, other receptors have been implicated in binding spike protein and promoting cell fusion. These receptors include CD147, AXL, KREMEN1, ASGR1, and α_5_β_1_ integrin. Dual cleavage of S protein by furin is also thought to be sufficient for initiation of the fusion process independent of ACE2 binding. **Dehydration/overcoming energy barriers**: Following S2′ cleavage by TMPRSS2 and dissociation of S1, the exposed fusion peptide inserts into the membrane of the target cell. Interaction with S protein triggers an increase in intracellular Ca^2+^ leading to activation of Ca^2+^-dependent ANO6 (TMEM16F) scramblase, which serves both as an ion channel and phospholipid scramblase. Notably, an increase in anionic phosphatidylserine externalization facilitates membrane fusion. Phosphatidylserine may draw out positively charged ions and regions of the fusion peptide, thereby neutralizing repulsive negative charges that limit initial plasma membrane contact. Folding back of the fusion peptide into a lower energy post-fusion state further helps to lower energy barriers to fusion. **Hemifusion Intermediate and Pore Expansion**: As the HR1-HR2 bundle folds and brings the two membranes together, formation of the hemi-fusion intermediate is stabilized by cholesterol. Cholesterol subsequently promotes lipid mixing and fusion pore expansion. **Fusion**: Fusion kinetics involving both host and viral factors result in the formation of SARS-CoV-2-infected multinucleated syncytia. Created with https://BioRender.com.

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
