# Peer review of "Mechanisms of Cell–Cell Fusion in SARS-CoV-2: An Evolving Strategy for Transmission and Immune Evasion"

_viruses, 2025, doi:10.3390/v17111405_

Round 1

Reviewer 1 Report

Comments and Suggestions for Authors

Chiang et al has delivered an informative review on the SARS-CoV-2 cell-cell fusion mechanism and its related factors influencing the process. The review is informative, but leaned heavier towards descriptive writing describing the process, and less so on the what the title implied on its contribution to pathogenicity and transmissibility. I have the following minor comments for the authors’ consideration.

Minor comments

Page 1 – Title: Majority of the review is on describing how cell-cell fusion and syncytia occurs in SARS-CoV-2’s infection. Factors influencing cell-cell fusion to happen over normal virion release and re-entry for dissemination were described in a briefer manner. Suggest changing the title to be more reflective of what is being described in the review

Page 1 – Title (and section 9): Similarly, how the cell-cell fusion and syncytia contribute to pathogenicity and transmissibility, is only mentioned in the final sections of the review. From the title I was expecting to read more on the clinical implications of fusogenic infection in transmission and disease severity, and how this may help in future management of infections with syncytia, especially in section 9.

Page 8 – In the following sentence “While syncytium formation may not strictly depend on furin- or TMPRSS2-mediated cleavage,39 cleavage at these sites is a critical step and may predict the fusogenic potential of emerging variants.”, as multiple types of cleavage sites were introduced in this paragraph’s writeup, can the author indicate clearly what “these sites” in the second part of the sentence referred to, to provide better clarify for the readers?

Page 12 – Section 5.2 Antibody mediate fusion paragraphs: can the author comment on whether these antibody mediate fusion can contribute to ADE of the SARS-CoV-2 infection? And whether there are any such findings in the literature?  

Page 13 – Section 6.1 and 6.2 Can the authors comment briefly on the implications of differential host factors/environment in a clinical context? If these differences can contribute to differential severity in a patient (e.g. how a febrile patient with higher increase in temperature compare with patient with minimal/no increase in temperature, with cell-cell fusion’s context)?

Reviewer 2 Report

Comments and Suggestions for Authors

The review article covered an important area of cell entry and cell-cell fusion for SARS CoV-2. The article is nicely put together and the topic is of broad interest to the readers of the journal.

Here are the improvements needed for the manuscript:

  • The title of the review is confusing. As the author pointed out that cell-cell fusion has only been implicated to pathogenicity but the title gives the impression that there is a direct link for the two. A more proper title should be “between transmissibility and immune evasion” as cell-cell fusion is shown to be involved in evading antibody neutralization. Transmissibility and immune escape are all from virus’s perspective, while pathogenicity is from the host perspective.
  • The authors should add more to the topics of how cell syncytia evade cellular responses as multinucleated cells are not efficiently cleared in severe COVID patients.
  • The author should clarify whether antibody mediated or enhanced transmission has been observed in human for SARS-CoV-2. This was a major concern for vaccination at the beginning of the pandemic due to the risk of antibody-dependent enhancement. “Antibody-mediated SARS-CoV-2 infection was observed in monocyte-derived macrophages and THP-1 cells.102 Anti-RBD IgG1 mAbs were shown to enhance THP-1 infection comparable to or even exceeding levels observed in permissive HeLa-ACE2 cells.102”. These are all in vitro cell experiments.

Minor edits:

  • Table 1 to use row separation line or space for easy to read, title should be Comparison of S dependent viral cell entry and cell-cell fusion.

Table 1, last line:  Integrins   Does not play ….., should be Do not play…..

Table 1 column title Fusion mechanism should be Factors or Fusion Effectors

  • I do not think furin counts as entry receptor, should be entry mediator: “Although furin is thought to serve as an entry receptor in the initiation of cell-cell fusion by efficiently cleaving S,103 binding of SARS-CoV-2 S to other membrane proteins have been reported to contribute to virus infection,104 but their role in cell-cell fusion remains to be explored.”
  • Virus infects but not transfects cells: “Multinucleated 293T-ACE2 cells transfected with SARS-CoV-2 readily internalized CCRF-CEM human T cell leukemia lymphoblasts expressing mCherry as well as other leukocyte cell lines including THP-1 monocytes, Jurkat T lymphocytes, Raji B cells, K562 human erythroleukemia cells, and PBMCs.80 Interestingly, internalization and cell death preferentially affected CD8+ over CD4+PBMCs.80”
  • Reference 5 has no mention of 6HB and fusion pore, should cite other papers for the following. “The fusion peptide maintains a high mechanical stability as the heptapeptide repeat 1 (HR1) and HR2 of the S2 subunit fold back to form 6HB.5 Subsequent formation of a fusion pore enables release of the ribonucleoprotein complex into the host cell.5”

Round 2

Reviewer 2 Report

Comments and Suggestions for Authors

All of my concerns are addressed in the new version.